# Performance and Mechanism of Nanoporous Ni@NiO Composites for RhB Ultrahigh Electro-Catalytic Degradation

Xiaoyu Wang [1], Fengda Pan [1], Xinhao Sun [1], Yongyan Li [1,2], Jun Zhou [1,*], Zhifeng Wang [1,2] and Chunling Qin [1,2,*]

1 School of Materials Science and Engineering, Hebei University of Technology, Tianjin 300401, China
2 Key Laboratory for New Type of Functional Materials in Hebei Province, Hebei University of Technology, Tianjin 300401, China
* Correspondence: zhoujun@hebut.edu.cn (J.Z.); clqin@hebut.edu.cn (C.Q.)

**Abstract:** Today, the development of new self-supporting electrode materials with high porosity and excellent degradation properties is of great importance for the removal of dye pollutants. Herein, this work synthesized nanoporous nickel@nickel oxide (np-Ni@NiO) electrode containing an amorphous alloy in the middle interlayer. The nanoporous structure endowed the electrode with more active sites and facilitated the ion/electron transport. The electrochemical active surface area was about 185.5 cm$^2$. The electrochemical degradation of rhodamine B (RhB) using a np-Ni@NiO electrode was systematically investigated. The effects of technology paraments (NaCl concentration, the applied potential and pH) on electro-catalytic degradation were explored. An RhB removal rate of 99.68% was achieved in 30 s at optimized conditions, which was attributed to the unique bicontinuous ligament/pore structure and more active sites on the surface, as well as lower charge transfer resistance. In addition, the degradation mechanism of RhB in electrochemical oxidation was proposed, according to active species capture tests and EPR measurements.

**Keywords:** nanoporous; Ni@NiO electrode; electrochemical oxidation; degradation

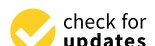



## 1. Introduction

Water pollution has increasingly become a serious environmental problem in modern industry. Persistent Organic Pollutants (POPs) [1] in wastewater are difficult to degrade in the treatment process due to their high resistance to degradation/decomposition.

The electrochemical advance oxidation processes (EAOPs) [2–7], owing to their higher catalytic efficiency than photocatalysis [8,9], especially in the mineralization of pollutants, have received increasing interest. The main electrochemical treatments used for wastewater treatment are electrocoagulation (EC) [10–12], electroreduction (ER) [13–15] and electrochemical oxidation (EO) [16–19], etc. Among them, electrochemical oxidation (EO) is the most commonly used electrochemical technology in EAOPs. EO produces indirect or mediated oxidation through active species generated by discharge on the anode surface. For example, physically adsorbed hydroxyl radical (M[·OH]) and "active chlorine" ($Cl_2$, HClO, $ClO^-$) species [20] were generated by electrolyte oxidation. A higher concentration of NaCl improved the removal efficiency of imidacloprid by the Ti/$PbO_2$ anode [21]. Hence, it is significant to investigate the effect of chloride ions on the mechanism of electrochemical degradation.

On the other hand, the electrode material plays an important role in the degradation activity. In recent studies, titanium mesh has usually been used as substrate to dope highly active catalyst [22–24]. However, the np-Ni@NiO prepared in this study is self-supporting and the preparation method is very simple. Most importantly, the np-Ni@NiO electrode, as a new substrate for replacing titanium mesh, had a promising application prospect in electrochemical oxidation. So far, most research has focused on precious metals and their

oxides, such as Ti/IrO$_2$ [25], Pt [26], Ti/Pt [27], Au [28], and Ti/RuO$_2$ [29], among others. While Ni metal and its oxides have been rarely studied for electrochemical degradation properties. Amorphous alloys as novel electrodes with a chemically homogeneous structure, which exhibit a good combination of excellent physicochemical properties, have attracted desirable attention in dye degradation. The active surface area and degradation efficiency can be further improved by chemically etching amorphous alloys for the preparation of heterogeneous nanostructures with highly specific surface areas.

In this work, the as-spun Ni$_{40}$Ti$_{40}$Zr$_{20}$ amorphous strips were immersed in a 0.05 M HF solution to obtain a sandwich-type thin strip with an amorphous alloy in the middle interlayer flanked by Ni@NiO. The np-Ni@NiO electrode composites were flexible and self-supporting.

The degradation performance of the np-Ni@NiO electrodes has been examined in the organic dye solutions. It was found that the electrode exhibited ultrahigh degradation performance. For example, it was reported that the Ti/RuO$_2$-IrO$_2$ electrode took 90 min to remove 100 ml of 50 mg/L RhB [30], while the gold nanoparticles took 50 min to remove 40 mL of 5 mg/L RhB [28]. Remarkably, the np-Ni@NiO electrodes prepared in this study required only 30 s to remove 40 mL of 10 mg/L RhB. In addition, the main influencing factors (including NaCl concentration, applied potential, pH, and dye species) for the degradation performance of np-Ni@NiO electrodes were systematically investigated. Then, active species capture experiments, as well as EPR measurements were carried out for exploring the degradation mechanism. The dealloyed Ni$_{40}$Ti$_{40}$Zr$_{20}$ amorphous thin strip has been used in previous studies for glucose sensors [31] and capacitor energy [32] storage. This study is the first time this material has been applied to the electrochemical degradation of dyes. It is discovered that the np-Ni@NiO electrodes demonstrate an excellent electrochemical degradation performance. This work broadens the scope for the application of amorphous alloys and nanoporous metals.

## 2. Materials and Methods

### 2.1. Preparation of np-Ni@NiO Electrode

The preparation method was as reported in a previous study [32]. The Ni, Zr and Ti metals with high purity (99.99%) were melted into alloy ingots by an electric arc furnace. The as-spun Ni$_{40}$Ti$_{40}$Zr$_{20}$ amorphous strips of 2 mm in width were obtained by the single-roller melt-spinning method. Finally, the as-spun samples were immersed in 0.05 M HF solution for 2, 4, and 6 h at 298 K. It was noted that the as-dealloyed strips for dealloying 2, 4, and 6 h were labeled as np-Ni@NiO-2, np-Ni@NiO-4, and np-Ni@NiO-6, respectively.

### 2.2. Microstructure Characterization

The crystal structure of the samples was examined by X-ray diffraction (XRD, Bruker, D8 Advance, Karlsruhe, Germany), the power of the XRD instrument was 4 kW, the test angle range was 10–90°, and the speed was 8° min$^{-1}$. The surface morphology of the catalysts was observed using scanning electron microscopy (SEM, FEI, Quanta 450 FEG, Hillsboro, OR, USA). Transmission electron microscopy (TEM, JEOL, JEM-2010, Akishima-shi, Japan) was used to examine the microstructure and elemental distribution of the samples. The constituent elements and chemical valences on the sample surface were characterized by X-ray photoelectron spectrometer (XPS, Thermo Fisher Scientific, ESCALAB 250Xi, Waltham, MA, USA). The passing energy and step size were 20 eV and 0.1 eV, respectively, while the charge correction was conducted with C 1 s = 284.8 eV binding energy as the standard energy.

### 2.3. Electrochemical Measurements

All electrochemical measurements were performed using an electrochemical workstation (Chenhua, CHI660E, Shanghai, China). Linear sweep voltammetry (LSV), cyclic voltammetry (CV), and electrochemical impedance spectroscopy (EIS) measurements were conducted in a three-electrode system. The prepared sample, a commercial Pt mesh and

a Ag/AgCl standard electrode were used as the working electrode, the counter electrode and the reference electrode, respectively.

*2.4. Electrochemical Degradation Measurements*

The 60 mL RhB solution (10 mg L$^{-1}$) containing 0.01 M NaCl was respectively placed into two 100 mL beakers for electrolysis. Using three electrode system, the np-Ni@NiO was used as the working electrode (the effective area is 1 cm$^2$). The two beakers were connected using a KCl salt bridge and the electrolytic cell was stirred during degradation using a magnetic stirrer.

The UV–visible diffuse reflectance spectra were recorded by UV–Vis spectrophotometer (Lengguang, UV 1920, Shanghai, China). The characteristic absorption peaks for rhodamine B (RhB), methyl orange (MO), acid orange II (AO II) and methylene blue (MB) were at 554, 464, 485 and 669 nm, respectively.

The catalytic activity of the np-Ni@NiO electrodes for pollutant dyes degradation (RhB, MO, AO II, MB) was examined under different conditions, including NaCl electrolyte concentration (0.01–0.1 M), applied potential (1.0–2.0 V), and pH (1, 7 and 14). To further study the reaction mechanism, 2 mL of 100 mM tert-butanol (TBA), methanol (MeOH), ammonium fluoride (NH$_4$F), and p-benzoquinone (PBQ) were added into the reaction solution as capture agents for active species. Bruker A300 electron paramagnetic resonance (EPR) was used to explore active radicals. Then, 5,5-dimethyl-1-pyrroline n-oxide (DMPO) was used to detect hydroxyl radicals (·OH) and superoxide radicals (·O$^{2-}$), while 2,2,6,6-tetramethyl-1-piperidinyloxy (TEMPO) was used for detecting single oxygen ($^1$O$_2$).

## 3. Results and Discussion

*3.1. Microstructure of Self-Supporting np-Ni@NiO Composite Electrodes*

Figure 1a shows the XRD patterns of the as-spun Ni$_{40}$Zr$_{20}$Ti$_{40}$ sample and as-dealloyed samples (np-Ni@NiO-2, np-Ni@NiO-4, np-Ni@NiO-6). No obvious diffraction peak is observed in the XRD pattern of the as-spun sample, indicating that the as-spun Ni$_{40}$Ti$_{40}$Zr$_{20}$ strip exhibits a homogeneous single amorphous phase. After dealloying, the new peak at ~44.5° is ascribed to the (1 1 1) crystal plane of fcc Ni (JCPDS 04–0850). The results demonstrate that the appearance of Ni results from the preferential dissolution of Zr and Ti elements, which is consistent with the previous study [33]. In addition, the amount of NiO may be too low to be detected.

The morphology evolution of the as-prepared samples for different dealloying time in 0.05 M HF solution was examined. Figure S1 shows that the amorphous precursor possesses a flat and simple surface. Figure 1b−d displays the surface morphologies of the as-prepared strip samples dealloyed in HF for 2, 4 and 6 h (np-Ni@NiO-2, np-Ni@NiO-4, np-Ni@NiO-6). The np-Ni@NiO-4 exhibits a bicontinuous and interpenetrating ligament/pore structure that is also called a spongy-like structure (Figure 1c), while the nanopores in the np-Ni@NiO-2 (Figure 1b) are discontinuous. As the dealloying treatment increases to 6 h (Figure 1d), the size of nanopores becomes enlarged due to the break of ligaments. Therefore, dealloying for 4 h is the most suitable process parameter.

Figure 1e shows the TEM image of np-Ni@NiO-4, where the bright and dark areas indicate nanoporous ligaments and pores, respectively. Figure 1f shows an HRTEM image of np-Ni@NiO-4. The lattice spacing of 1.76 nm in region A is attributed to the (2 0 0) crystal plane of Ni (JCPDS #04-0850), while the lattice spacing of 2.08 nm in region B is attributed to the (2 0 0) crystal plane of NiO (JCPDS #04-0835). The results indicate that np-Ni@NiO-4 surface consists of Ni and NiO. A typical core@shell (Ni@NiO) structure is formed on the surface of the dealloyed strip by the dealloying-natural oxidation process.

Moreover, in order to clarify the elemental information of the as-spun and np-Ni@NiO-4 strips, XPS measurements were carried out and the obtained data was fitted by Gaussian method (Figure 2). As shown in Figure 2a,b, Zr 3d spectrum in the as-spun samples was composed of Zr$^0$ and Zr$^{4+}$, while Ti 2p spectrum in as-spun was composed of Ti$^0$, Ti$^{2+}$, Ti$^{3+}$, and Ti$^{4+}$ [32]. After reacting with HF, the detection signals of Zr 3d and Ti 2p for np-

Ni@NiO can hardly been observed. As shown in Figure 2c, the as-spun sample exhibits $Ni^0$ species centered at 852.2 eV and 869.5 eV. For the np-Ni@NiO sample, the peaks centered at 855.5 and 873.2 eV can be assigned to the characteristic peaks of $Ni^{2+}$. Whereas the peaks located at 861.0 and 879.1 eV represent the satellite peaks attributed to $Ni^{2+}$ $2p_{3/2}$ and $Ni^{2+}$ $2p_{1/2}$ [31], which originates from the shake-up of the $Ni^{2+}$ $2p_{3/2}$ and $Ni^{2+}$ $2p_{1/2}$ edge at the high binding energy sides, respectively.

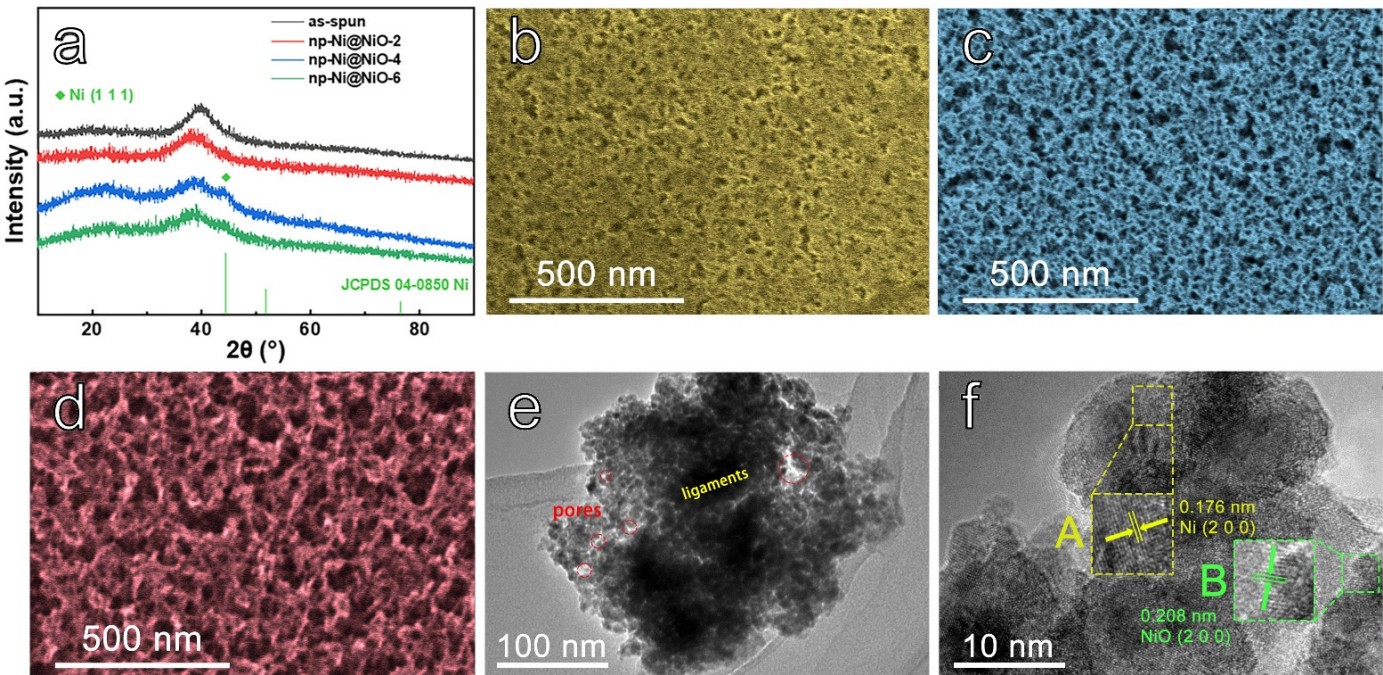

**Figure 1.** (**a**) XRD patterns of as-spun $Ni_{40}Ti_{40}Zr_{20}$ and as-dealloyed samples. SEM images of the as-spun samples immersed in 0.05 M HF at 298 K for (**b**) 2 h; (**c**) 4 h and (**d**) 6 h. (**e**) TEM image and (**f**) corresponding HRTEM image of np-Ni@NiO-4.

Combining the XPS analysis with XRD, SEM and TEM examinations, it can be concluded that the nanoporous Ni@NiO (np-Ni@NiO) electrodes are successfully synthesized via dealloying as-spun $Ni_{40}Ti_{40}Zr_{20}$ amorphous strips.

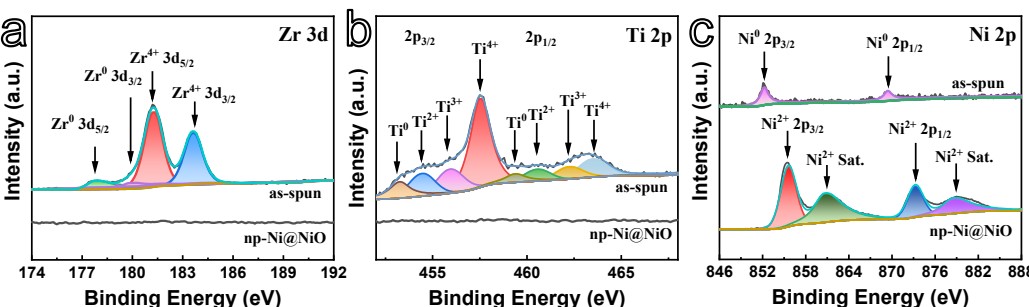

**Figure 2.** (**a**) Zr 3d XPS spectra; (**b**) Ti 2p XPS spectra; (**c**) Ni 2p XPS spectra for the as-spun $Ni_{40}Ti_{40}Zr_{20}$ and the np-Ni@NiO-4.

### 3.2. Electrochemical Measurements

The electrochemical degradation performance of np-Ni@NiO electrodes are shown in Figure 3a. In addition, the degradation performance of the as-spun sample is also examined for comparison. The degradation performance of the electrodes could be speculated from the dye concentration according to $C_t/C_0$ ($C_t$ and $C_0$ are the RhB concentrations at time t

and the original state, respectively). It is found that the as-spun sample shows negligible degradation performance. After dealloying, the as-dealloyed samples (np-Ni@NiO) exhibit remarkably high degradation rates. The degradation time for completely degrading RhB by the np-Ni@NiO-4 electrode is as fast as 30 s in the mixed RhB solution (as shown in Video S1). Figure 3b shows the typical UV absorption spectra of the mixed RhB solution. The characteristic absorption peak (RhB) at λ = 554 nm becomes flattened at 30 s. Meanwhile, the color of the RhB solution becomes transparent with degradation time, as shown in the inset of Figure 3b. Moreover, the np-Ni@NiO-4 also demonstrates extremely high reusability in Figure S2. After 10 cyclic electrocatalytic measurements, the degradation performance of np-Ni@NiO-4 consistently achieved over 99%.

The electrochemically active surface area (ECSA) could be estimated by the electrochemical double layer capacitance ($C_{dl}$) method [34]. Figure S3 shows the CV curves of np-Ni@NiO electrode in 0.1 M KOH at different scan rates in the potential range of 0–0.1 V. Then, the current density of np-Ni@NiO and the as-spun samples obtained from CV curves are showed in Figure 3c. As can be seen from Figure 3c, the slope ($C_{dl}$) of the np-Ni@NiO electrodes is much larger than that of the as-spun sample. Moreover, the values of $C_{dl}$ for np-Ni@NiO-2, np-Ni@NiO-4 and np-Ni@NiO-6 are 6.02, 7.42 and 5.96 $\mu F \ cm^{-2}$, respectively. Then, the ESCA values of np-Ni@NiO are derived as 150.5 $cm^2$ for np-Ni@NiO-2, 185.5 $cm^2$ for np-Ni@NiO-4 and 149 $cm^2$ for np-Ni@NiO-6 [35–37]. It reveals that the ECSA of the as-prepared samples utilized from the dealloying treatment are significantly increased. Such high ECSA provides fertile active sites, promotes the adsorption of dye molecules, and, thus, accelerates the decomposition of dye molecules. Catalytic activity of electrodes were measured in Figure S4, these results reveal that the existence of nanopores effectively enhances the electrocatalytic activity of materials. This result is consistent with ECSA.

It is well known that, for the degradation of organic dyes, oxygen evolution reaction (OER) is the main side reaction. The higher the oxygen evolution potential of the anode means that more current is used for electrogenerated active species. LSV curves and Tafel plots (Figure 3d,e) were used to evaluate the oxygen evolution activity of electrode materials in 1 M KOH. Clearly, the oxygen evolution overpotential for np-Ni@NiO-4 (45.7 mV) was slightly higher than those of np-Ni@NiO-2 (39.0 mV) and np-Ni@NiO-6 (42.1 mV) (Figure 3d). Moreover, np-Ni@NiO-4 has the highest Tafel slope among the as-dealloyed samples (Figure 3e). Therefore, the np-Ni@NiO-4 electrode could efficaciously suppress oxygen evolution side reactions, thereby improving current efficiency.

The electron transfer dynamics of electrodes in electrochemical oxidation were studied by electrochemical impedance spectroscopy (EIS) [38]. Figure 3f shows the Nyquist plots of the samples. The semicircle under the high-frequency region represents the charge transfer resistance of the electrode materials, while the slash under the low-frequency region represents the ion diffusion resistance of the electrochemical reaction. Although the as-dealloyed samples (np-Ni@NiO-2, np-Ni@NiO-4, and np-Ni@NiO-6) possess almost a similar charge transfer resistance, the semicircles in the high frequencies for the as-dealloyed samples largely decrease compared to that of the as-spun electrode, indicating that the nanostructure modification is effective in decreasing the charge transfer resistance. Meanwhile, at low frequencies, the np-Ni@NiO-6 electrode shows the largest slope, demonstrating that it has the least ions diffusion resistance. The reason could be that the enlargement of the pores facilitates the diffusion of ions.

### 3.3. Electrochemical Degradation Performance of np-Ni@NiO

The technique parameters (NaCl concentration, applied potential and pH value) greatly influence the electrochemical degradation performance of the np-Ni@NiO electrode. Then, the degradability of np-Ni@NiO-4 was studied at different NaCl concentrations (0.01–0.5 M), applied potentials (1–2 V) and pH values (1–14), as shown in Figure 4. In addition, the versatility of the np-Ni@NiO-4 electrode for various dyes were also studied (Figure 5).

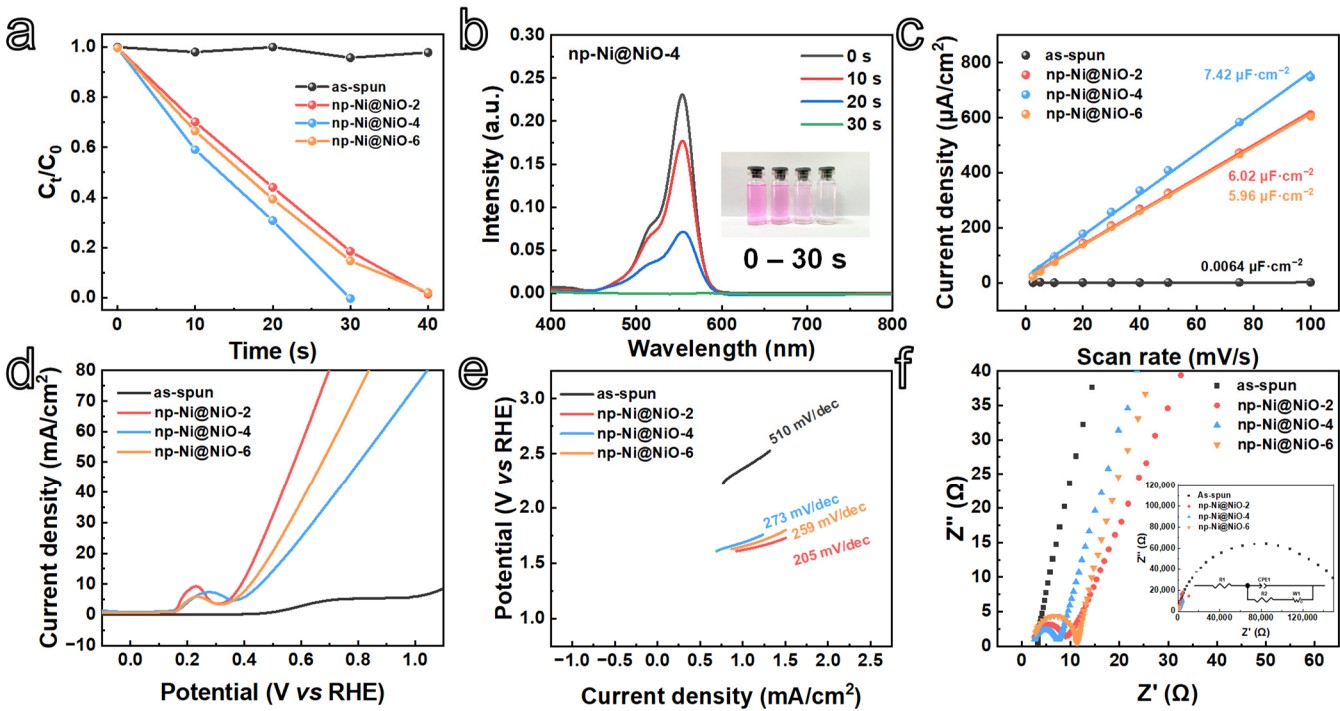

**Figure 3.** (**a**) The electrochemical degradation of RhB for different catalysts (0.01 M NaCl concentration, 1.5 V applied potential and initial pH 7). (**b**) UV absorption spectra of RhB solutions under various times at np-Ni@NiO-4 electrode. (**c**) Current density differences at 0.05 V (vs Ag/AgCl) plotted against the scan rate measured in a non-Faradaic range. (**d**) LSV curves measured at 5 mV s$^{-1}$ in 1 M KOH. (**e**) Tafel plots derived from LSV curves. (**f**) The Nyquist plots of the electrode materials in 0.5 M Na$_2$SO$_4$.

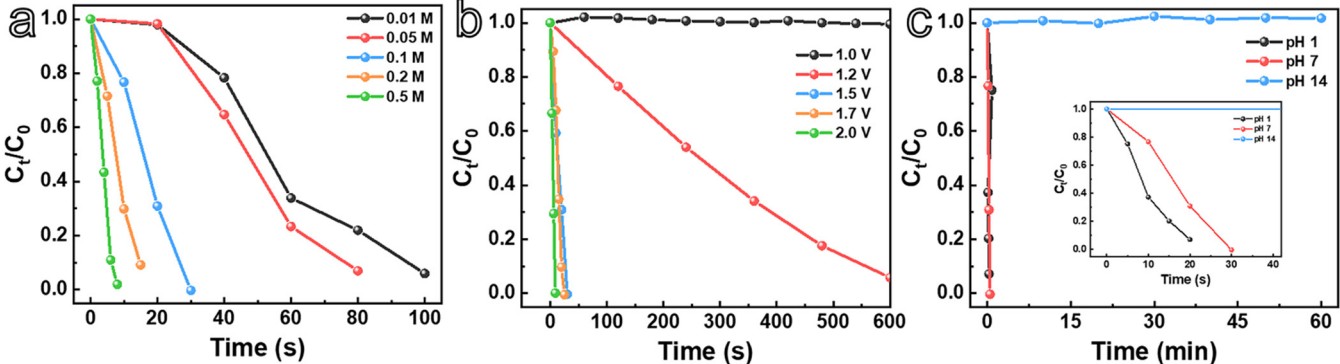

**Figure 4.** (**a**) The effect of the NaCl concentration on degradation performances of np-Ni@NiO-4 (1.5 V applied potential, initial pH 7). (**b**) The effect of the applied potential on degradation performances of np-Ni@NiO-4 (0.1 M NaCl concentration, initial pH 7). (**c**) The effect of the pH on degradation performances of np-Ni@NiO-4 (0.1 M NaCl concentration, 1.5 V applied potential).

### 3.3.1. The Effect of Chlorine Ions Concentration

The addition of chloride ions promotes the formation of strong oxidants (active chlorine including Cl$_2$, HClO, and OCl$^-$), which help to degrade RhB. The suitable concentration of chloride ions not only improves the degradation performance of the electrode material but also reduces its energy consumption. The effect of NaCl concentration on the electrochemical oxidative degradation of RhB is shown in Figure 4a. The results show that the degradation performance of np-Ni@NiO-4 increases with an increase in NaCl concentration. When the NaCl concentration reaches 0.05 M, the np-Ni@NiO-4 electrode

takes 80 s to remove the RhB. As the NaCl concentration further increases to 0.1 M or more, only 30 s or less is required. On the other hand, an increase in NaCl concentration could reduce the discharge potential of active chlorine ($Cl_2$, $HClO$, $OCl^-$), and more current is consumed [39]. Thus, the current consumption also needs to be considered. The consumed current is calculated via Equation (1).

$$W = U \int IT \tag{1}$$

where W is the consumed electric work, U is the applied potential, I is the current and T is the time to completely degrade RhB. Figure S5 displays the chronoamperometric response of np-Ni@NiO electrodes in the mixed RhB solution with a NaCl concentration from 0.01–0.5 M. The current consumption for 0.1 M NaCl is minimal and is calculated to be 0.127 J. Therefore, the NaCl concentration is optimized to be 0.1 M.

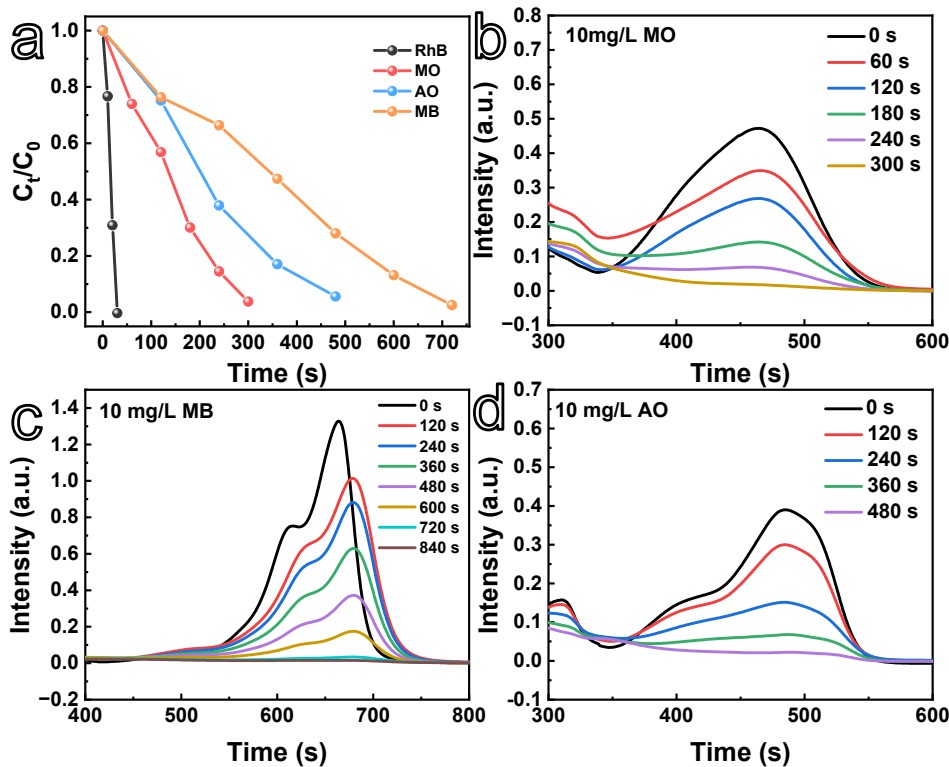

**Figure 5.** The typical UV absorption spectra of the (**a**) MO; (**b**) AO; (**c**) MB by the np-Ni@NiO-4 electrode for different times. (**d**) Degradation rates of different dyes versus time.

### 3.3.2. The Effect of Applied Potential

Figure 4b shows the effect of applied potential on the degradation performance of the np-Ni@NiO-4 electrode. The results show that the degradation performance is closely dependent on the applied potential. At an applied potential of 1 V, almost no degradation behavior occurs (Figure 4b). When the applied potential increases to 1.2 V, a visible degradation behavior of the np-Ni@NiO-4 electrode can be observed. Further increasing the applied potential to 1.5–2.0 V, the degradation performance of the np-Ni@NiO-4 electrode is tremendously enhanced. The optimized potential for electrochemical oxidative degradation of RhB is determined to be 1.5 V.

### 3.3.3. The Effect of pH

The effect of pH on the degradability of np-Ni@NiO-4 was explored. Figure 4c shows the influence of pH on the degradation of np-Ni@NiO-4. The np-Ni@NiO-4 exhibits

negligible degradation performance in solution at pH 14. When the pH value is 1 and 7, the degradation performance of np-Ni@NiO-4 is noticeably improved.

The influence of pH on the degradability of np-Ni@NiO-4 can be attributed to two factors: the oxygen evolution side reaction and the relative concentrations of active chlorine species. The acidic solution can increase oxygen overpotential and improve the formation of ·OH [40–42]. It is well known for active chlorine ($Cl_2$, $HClO$, $ClO^-$), that $Cl_2$ acts as the dominant species at pH < 3, $HClO$ in the pH range from 3–8; and $ClO^-$ at pH > 8 [43]. The oxidation of pollutants will be facilitated in acidic environments (pH 1) due to the higher oxidation potential of $Cl_2$ ($E^0$ = 1.36 V vs. SHE) and $HClO$ ($E^0$ = 1.63 V vs. SHE) compared to $ClO^-$ ($E^0$ = 0.89 V vs. SHE) [44].

Therefore, considering the maintenance of the equipment and the reusability of the electrodes in acidic solutions, the initial solution pH of 7 is the optimum condition for the treatment of RhB solution.

### 3.3.4. The Versatility of the np-Ni@NiO Electrode for the Degradation of Different Organic Dyes

To examine the versatility of the np-Ni@NiO electrode for the degradation of different dyes, three typical organic dyes including methyl orange (MO), acid orange II (AO) and methyl blue (MB), were used as simulated pollutants.

Figure 5a–c shows the change in the UV–vis absorption spectra of MO, AO, and MB with degradation time. The characteristic peaks of MO, AO, and MB are 464, 485, and 669 nm, respectively. The MO, AO and MB dyes degraded by the electrochemical oxidation occur at around 5, 8 and 12 min, respectively, indicating the np-Ni@NiO-4 electrode exhibits high electro-oxidation degradation performance for MO, AO, and MB. It also indicates that the np-Ni@NiO-4 can be used as a highly efficient electro-oxidation catalyst for degrading various types of organic dyes.

Table 1 displays a comparison of electrochemical degradation performance among different materials. It can be seen that the np-Ni@NiO composite prepared in this work has ultrahigh degradability in terms of dye removal, particularly RhB. In addition, the preparation process of the np-Ni@NiO-4 composites is simple and cost-effective compared with those of other composite electrodes. More importantly, the electrode has a super-efficient degradation performance without the addition of precious metals.

**Table 1.** Comparison of electrochemical degradability among different electrodes materials.

| Materials | Dyes | Degradation Efficiency | Electrochemical Conditions | Ref. |
|---|---|---|---|---|
| AuNPs | RhB (5 mg $L^{-1}$) | 50 min 100% | 28 mA $cm^{-2}$, 0.4 M KCl | [28] |
| Ti/$RuO_2$–$IrO_2$ | RhB (50 mg $L^{-1}$) | 90 min 100% | 40 mA $cm^{-2}$, 0.5 M $Na_2SO_4$ | [30] |
| $NiCo_2O_4$ | RhB (1000 mg $L^{-1}$) | 20 min 100% | 18 mA $cm^{-2}$, 0.05 M $Na_2SO_4$ | [45] |
| $RuO_2$–PdO–$TiO_2$/Ti | RhB ($1.043 \times 10^{-5}$ mol $L^{-1}$) | 30 min 90.4% | 20 mA $cm^{-2}$, 0.1 M $Na_2SO_4$ | [46] |
| Ti/$SnO_2$–Sb | RhB (50 mg $L^{-1}$) | 20 min 100% | 3.04 V, 0.05 M $Na_2SO_4$ | [47] |
| Ti/$TiRuO_2$ | MB (80 mg $L^{-1}$) | 150 min 97.3% | 20 mA $cm^{-2}$, 0.6 g $L^{-1}$ $Cl^-$ | [48] |
| $TiO_2$-NiO@Sb-$SnO_2$ | MB (5 mg $L^{-1}$) | 120 min 100% | 5 mA $cm^{-2}$, 0.5 M NaCl | [49] |
| Ti/$RuO_2$ | MO (50 mg $L^{-1}$) | 240 min 81% | 70 mA $cm^{-2}$, 0.05 M $Na_2SO_4$ | [50] |
| $TiRuSnO_2$ | MO (100 mg $L^{-1}$) | 3 h 75% | 10 mA $cm^{-2}$, 0.5 M $Na_2SO_4$ | [51] |
| np-Ni@NiO-4 | RhB (10 mg $L^{-1}$) | 30 s 99.68% | 1.5 V (or 6 mA $cm^{-2}$), 0.1 M NaCl | This Work |

### 3.4. Active Species Test

It is well known that the activated chlorine ($Cl_2$, $HClO$, $ClO^-$), hydroxyl radicals (·OH) and superoxide radicals (·$O_2^-$) are involved in the process of electrochemical oxidation. To further explore the effect of active species on degradation efficiency during the electro-oxidation process, active species capture tests (Figure 6a,b) and EPR (Figure 6d–f) measurements were carried out.

In active species capture experiments, tert-butanol (TBA) acts as a scavenger for $\cdot$OH and active chlorine ($Cl_2$, HClO, $ClO^-$). In addition, methanol (MeOH), ammonium fluoride ($NH_4F$), and 1,4-benzoquinone (PBQ) are added to the mixed RhB solution to capture $\cdot$OH, HClO and $\cdot O_2^-$, respectively. The degradation efficiency without scavengers (referred to as blank) is plotted in Figure 6a,b for comparison.

The addition of MeOH and PBQ for capturing active species $\cdot$OH and $\cdot O_2^-$, respectively, resulted in a reduction in degradation efficiency from 99.68% to 64.83% and 51.63% at 30 s, respectively. With the addition of TBA to capture the active chlorine and $\cdot$OH, the degradation efficiency is greatly suppressed from 99.68% to 14.63%. Apparently, active chlorine ($Cl_2$, HClO, $ClO^-$) plays a key role in the degradation process. Furthermore, the presence of $NH^{4+}$ for capturing HClO has a significant inhibitory effect on the degradation of RhB; the degradation efficiency decreases from 99.68% to 6.56%. From Figure 6a,b, it is clearly seen that the HClO species acts as the dominating active species among all of the active species. Therefore, the effect of different active species on the removal rate of RhB is as follows: activated chlorine (HClO being the predominant species) > $\cdot O_2^-$ > $\cdot$OH.

To further validate the active chlorine ($Cl_2$, HClO, $ClO^-$) produced in the electrochemical oxidation system. The DPD (N,N-diethyl-p-phenylenediamine) spectrophotometric method was used to measure the active chlorine [52]. Figure 6c shows the absorbance curve of the 0.1 M NaCl solution with the addition of DPD plotted with different times. The peak intensity is proportional to the active chlorine concentration. As shown in Figure 6c, the peak at $\lambda$ = 515 nm increases as the electrolysis proceeds, which proves that active chlorine ($Cl_2$, HClO, $ClO^-$) is continually being produced in the reaction.

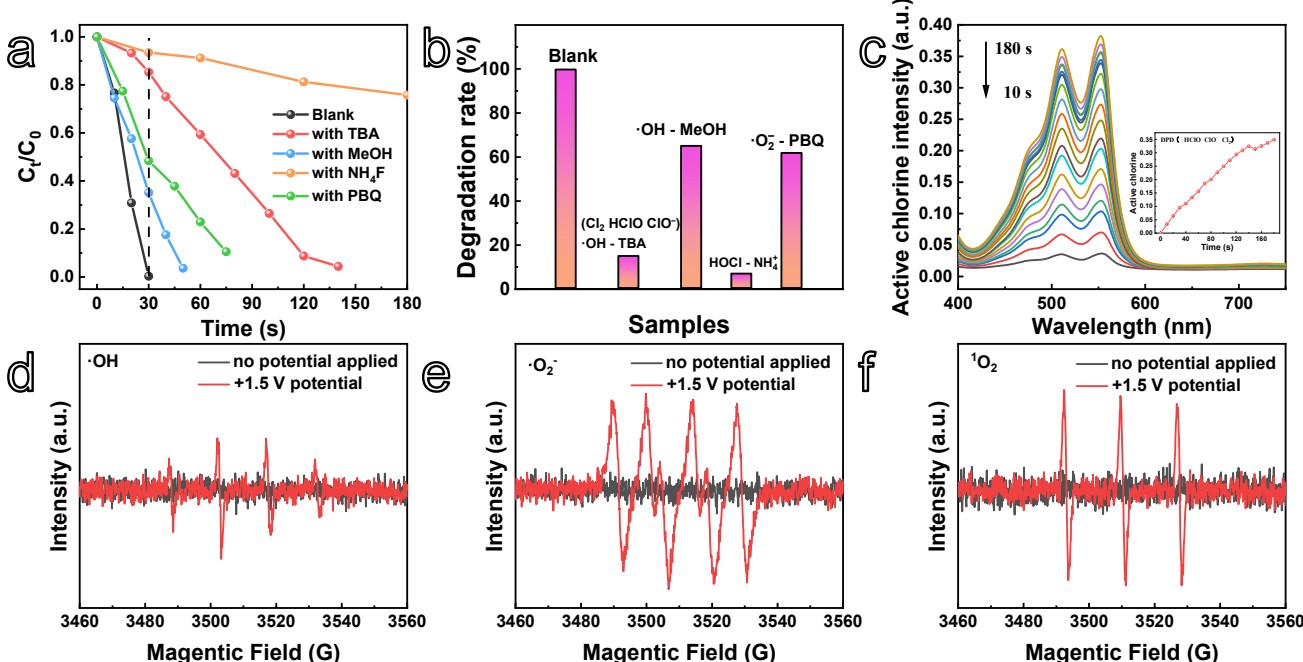

**Figure 6.** (**a**) Relative concentration of RhB versus time after the addition of scavenger. (**b**) The effect of scavengers on the degradability of np-Ni@NiO-4. (**c**) The absorbance curve of the NaCl solutions under different times with the addition of DPD (the inset shows the peak intensity at $\lambda$ = 515 nm curve versus time). The EPR spectra of (**d**) $\cdot$OH; (**e**) $\cdot O_2^-$; and (**f**) $^1O_2$ generated under different conditions for the np-Ni@NiO-4.

On the other hand, the generation of free radicals ($\cdot$OH, $\cdot O_2^-$, and $^1O_2$) in the electrochemical oxidation process was further examined by EPR measurements, as shown in Figure 6d–f. The generation of $\cdot$OH radicals before and after the applied potential is shown in the EPR spectra (Figure 6d). For the reaction system under the applied potential condition, the $\cdot$OH radical peaks appear, which is due to the generation of $\cdot$OH adsorbed

on the electrode surface by the discharge of $H_2O$ at the anode. However, it seems that the amount of ·OH is low, and it is assumed that the generated ·OH underwent a rapid conversion to other active species ($·O_2^-$ and $^1O_2$) according to Equations (2)–(4). Then, the high $·O_2^-$ and $^1O_2$ peaks for the electrochemical system under the applied potential are observed in Figure 6e,f, illustrating that a large amounts of $·O_2^-$ and $^1O_2$ were produced in the reaction system.

$$·OH + ·OH \ \rightarrow \ H_2O_2 \tag{2}$$

$$·OH + H_2O_2 \ \rightarrow \ ·O_2^- + H_2O \tag{3}$$

$$·OH + ·O_2^- \ \rightarrow \ ^1O_2 + OH^- \tag{4}$$

According to the combined analysis of active species capture experiments (Figure 6a,b), the DPD spectrophotometric method (Figure 6c), as well as EPR measurements (Figure 6c–d), a variety of active species ($·OH$, $·O_2^-$, $^1O_2$, $Cl_2$, $HClO$, $ClO^-$) are produced during the electrochemical oxidation process. Among the active species, active chlorine ($Cl_2$, $HClO$, $ClO^-$) plays a major role, while $·O_2^-$ and $^1O_2$, which are rapidly generated due to the existence of ·OH, act as secondary roles in the degradation process.

*3.5. Mechanism of Electro-Oxidation Degradation*

Electrochemical oxidation is applied as an electrochemically advanced oxidation process for the degradation of dyes. The degradation process is quite complex, since multiple active substances are produced. The removal efficiency of organic pollution is determined by the catalytic activity of the anode. The mechanism of removing dye molecules is proposed in Figure 7.

First, ·OH radicals are produced by the discharge of $H_2O$ and the ·OH radicals are further adsorbed on the np-Ni@NiO anode surface (S[·]) according to Equation (5). Then, the part of the ·OH radicals directly oxidize the pollutants (R) to $CO_2$ and $H_2O$ based on Equation (6).

$$S[\ ] + H_2O \ \rightarrow \ S[·OH] + H^+ + e^- \tag{5}$$

$$S[·OH] + R \ \rightarrow \ S[\ ] + RO \ (or \ CO_2) + H_2O \tag{6}$$

Meanwhile, based on Equations (3) and (4), the majority of ·OH radicals adsorbed on the electrode surface are rapidly converted to $·O_2^-$ and $^1O_2$ oxidants, accompanying the oxygen evolution side reactions. The generation of a large amount of $·O_2^-$ and $^1O_2$ with high oxidizability in the dyes could be helpful for degrading dye pollutants.

Moreover, as discussed in Figure 7, active chlorine ($Cl_2$, $HClO$, $ClO^-$) are produced on the surface of the electrode in the mixed dye solution containing NaCl. The yielding of the active chlorine ($Cl_2$, $HClO$, $ClO^-$) follows Equations (7)–(9). First, the $Cl^-$ cations are oxidized to $Cl_2$ on the surface of the anode by electro-oxidation. Then, the released $Cl_2$ reacts with $H_2O$ to produce $HClO$. Meanwhile, the $HClO$ species further dissociates into $ClO^-$ and $H^+$ in the bulk solution.

$$2Cl^- \ \rightarrow \ Cl_{2\ (aq)} + 2e^- \tag{7}$$

$$Cl_{2\ (aq)} + H_2O \ \rightarrow \ HClO + H^+ + Cl^- \tag{8}$$

$$HClO \ \rightarrow \ H^+ + ClO^- \tag{9}$$

Apparently, the amount of active species is closely dependent on the electro-oxidation catalytic activity of electrode materials. The fertile porous channels and more active material NiO sites of the present np-Ni@NiO electrode composites not only accelerate the rapid transmission of the electrolyte ions, but also facilitate the absorptance of ·OH on the surface of np-Ni@NiO, which results in producing more active species. Overall, the generation of a large amount of active chlorine ($Cl_2$, $HClO$, $ClO^-$) together with $·OH$, $·O_2^-$, and $^1O_2$ active species on the np-Ni@NiO surface by the electro-oxidation in the mixed dye solution containing NaCl could be responsible for the ultrafast degradation rate.

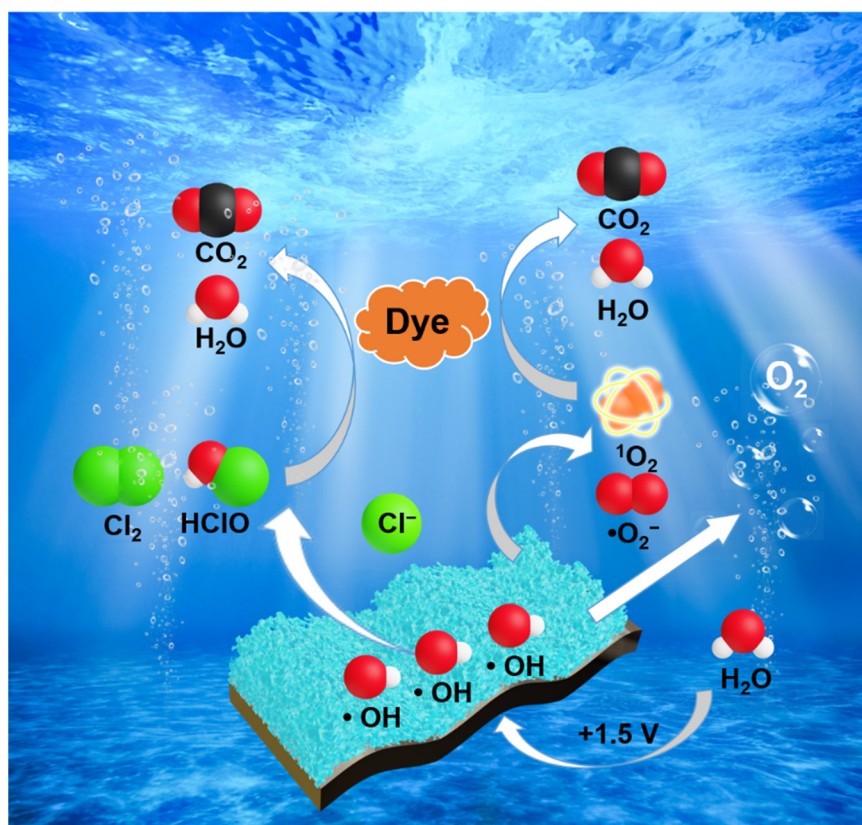

**Figure 7.** Schematic illustration of the electrocatalytic degradation of dye molecules by np-Ni@NiO-4 electrode.

## 4. Conclusions

In this work, a self-supporting np-Ni@NiO electrode was prepared by a simple dealloying method. The obtained electrode possesses unique 3D porous architecture, which endows the electrode with high electrochemical active surface area (185 cm$^2$). Moreover, diversified active species are generated via electro-oxidation in 0.1 NaCl solution at pH 7. The unique porous structure and the generation of fertile active sites for the electrode result in an ultrafast degradation rate (RhB degradation for 30 s) and excellent high reusability. Moreover, the ultrafast EO process is attributed to the generation of diversified active species ($\cdot OH$, $\cdot O_2^-$, $^1O_2$, $Cl_2$, $HClO$, $ClO^-$). Therefore, the newly self-supporting np-Ni@NiO as an electrocatalysis electrode will largely extend to commercial applications.

**Supplementary Materials:** The following supporting information can be downloaded at: https://www.mdpi.com/article/10.3390/met13010038/s1, Figure S1: SEM images of the as-spun sample; Figure S2: Reusability of np-Ni@NiO anode for electrochemical degradation of RhB; Figure S3: Cyclic voltammograms of as-spun (**a**), np-Ni@NiO-2 (**b**), np-Ni@NiO-4 (**c**) and np-Ni@NiO-6 (**d**) electrode in 0.1 M KOH at different scan rates in the potential range of 0–0.1 V; Figure S4: CV (sweep rate of 0.05 V/s) curves for electrode materials in 0.5 M $Na_2SO_4$ (the inset shows the value of voltammetric charge); Figure S5: The chronoamperometric response of np-Ni@NiO electrodes under a NaCl concentration from 0.05–0.1 M; Video S1: The electrochemical degradation of RhB for the np-Ni@NiO-4 (10 mg/L RhB, 0.1 M NaCl, 1.5 V applied potential, pH 7).

**Author Contributions:** Conceptualization, C.Q. and X.W.; methodology, X.W. and F.P.; validation, Y.L., Z.W. and J.Z.; formal analysis, X.W. and X.S.; investigation, X.W. and F.P.; resources, C.Q. and Z.W.; data curation, X.W. and Y.L.; writing—original draft preparation, X.W. and X.S.; writing—review and editing, Y.L., Z.W. and J.Z.; visualization, X.S.; supervision, C.Q. and J.Z.; project administration, C.Q.; funding acquisition, C.Q., Y.L. and Z.W. All authors have read and agreed to the published version of the manuscript.

**Funding:** This work is financially supported by the National Natural Science Foundation of China (52071125), Natural Science Foundation of Hebei Province, China (E2020202176; E2020202071), Science and Technology Project of Hebei Education Department, China (ZD2020177).

**Data Availability Statement:** The data presented in this study are available on request from the corresponding author.

**Conflicts of Interest:** The authors declare no conflict of interest.

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
