# Peer review of "Performance and Mechanism of Nanoporous Ni@NiO Composites for RhB Ultrahigh Electro-Catalytic Degradation"

_metals, doi:10.3390/met13010038_

Round 1

Reviewer 1 Report

This paper reports on the synthesis of Ni@NiO electrodes by a dealloying method starting from Ni-Ti-Zr strips, and on their subsequent characterization by means of various techniques, including electron microscopies, XRD and XPS. The system activity in the degradation of RhB is also investigated by optical analyses and electrochemical techniques.

Among the thousands of works on analogous topics, I do not understand where the novelty significance of this study lies. The work is presented as a limited quality technical report and suffers from many problems/drawbacks, that make it definitely unacceptable for publication. The main reasons for this decision are summarized in the following list:

1.      Many eco-friendly materials completely free from any environmental impact are already available for the target application, as the authors should know. Is the present one a meaningful exercise?

2.      In the Introduction, which is rather generic and contains trivial observations, the authors fail to demonstrate the importance and impact of the present study, that does not offer any unprecedented insight, breakthrough or conceptual leap which may represent a significant advancement with respect to the state-of-the-art in the field.

3.      Section 2.2 is too poor of details regarding the operating conditions used for the various analytical characterizations. For instance, in the case of XPS, nothing is told about the used charging correction procedure, pass energy, resolution and fitting details.

4.      Electron microscopy images in Fig. 1b-e and S1 are of a too low resolution and magnification to enable a really meaningful insight into the system morphology and organization. The image in Fig. 1f is of a very low quality and the writings preclude a real observation of the system nano-organization.

5.      Comments on XPS data (lines 150-160) are unacceptably poor and completely lacking from a detailed comparison with pertaining literature works. The fitting in Fig. 2b is highly questionable and the top spectrum in Fig. 2d is almost at the noise level.

6.      The supplementary video is almost useless.

7.      Sections 3.2 and 3.3 are very long-winded and not properly targeted to the presented experimental data.

8.      Fig. 3f: the equivalent circuit model used to fit the data is not reported.

9.      The role of chloride ions is not properly discussed and not adequately supported by chemical analyses on the working solutions to identify the real occurrence of the various species. In this regard, the mechanism section on page 11, is very speculative and not adequately supported, and Fig. 7 (which is also badly drawn) remains almost completely useless.

10.  No chemico-physical analyses on the working materials after functional tests are presented and discussed.

11.  The Conclusions are very technical and not adequately supported by the data, and no future perspectives for the prosecution of the presented research activities can indeed be foreseen.

Author Response

Responses to the reviewers

Reviewers' comments:

Reviewer #1: This paper reports on the synthesis of Ni@NiO electrodes by a dealloying method starting from Ni-Ti-Zr strips, and on their subsequent characterization by means of various techniques, including electron microscopies, XRD and XPS. The system activity in the degradation of RhB is also investigated by optical analyses and electrochemical techniques.

Among the thousands of works on analogous topics, I do not understand where the novelty significance of this study lies. The work is presented as a limited quality technical report and suffers from many problems/drawbacks, that make it definitely unacceptable for publication. The main reasons for this decision are summarized in the following list:

  1. Many eco-friendly materials completely free from any environmental impact are already available for the target application, as the authors should know. Is the present one a meaningful exercise?

>> First, the authors would like to thank the reviewer's valuable comments. We agree with you. Although many environmental-friendly materials have been prepared, we believe that materials developed in this work also demonstrated unique advantages such as porous structures, good electrical conductivity as well as self-supporting feature. (1) In the field of electrochemical degradation of organic pollutants, most studies have focused on Ti, Ru, Ir, Sn, Ta, Sb, Pb and boron-doped diamond. However, Ni metals and their oxides have rarely been investigated. (2) The dealloyed Ni40Ti40Zr20 amorphous strip has been applied in previous studies in the fields of glucose sensors and supercapacitor. In this study, it is the first time that this material is applied to the field of electrochemical degradation of dyes, which broadens the scope of application of amorphous alloys. Moreover, the material exhibits extremely fast degradation performance, which is worth to be explored. (3) In previous studies, many researchers have prepared the electrodes by doping catalysts on the surface of Ti mesh, such as Ti/Pt, Ti/RuxIryOz, Ti/IrO2-Ta2O5, Ti/SnO2-Sb. The preparation methods were very complex. By contrast, the preparation method of self-supporting np-Ni@NiO was simple. Moreover, the np-Ni@NiO can replace Ti mesh as a new self-supporting substrate to further coat and grow the active substances.

  1. In the Introduction, which is rather generic and contains trivial observations, the authors fail to demonstrate the importance and impact of the present study, that does not offer any unprecedented insight, breakthrough or conceptual leap which may represent a significant advancement with respect to the state-of-the-art in the field.

>> Thanks to the reviewer’s suggestion. We amended the Introduction and the modification was shown in the revised manuscript (Page 1,2).

  1. Section 2.2 is too poor of details regarding the operating conditions used for the various analytical characterizations. For instance, in the case of XPS, nothing is told about the used charging correction procedure, pass energy, resolution and fitting details.

>> Thanks to the reviewer's comments. Section 2.2 was optimized in the revision.

  1. Electron microscopy images in Fig. 1b-e and S1 are of a too low resolution and magnification to enable a really meaningful insight into the system morphology and organization. The image in Fig. 1f is of a very low quality and the writings preclude a real observation of the system nano-organization.

>> The Ni-based materials possess the magnetic properties, resulting in the fuzzy SEM images. Actually, from SEM (Fig. 1b-e and S1), it is clearly seen the nanopores and ligaments. The quality of images was similar to those of reported papers (Applied Materials Today 19 (2020) 100539; Journal of Power Sources 247 (2014) 896). We further enlarged the SEM images in the revised Fig. 1b-e. Moreover, owing to the coronavirus pandemic, TEM measurements are currently not available. The image in Fig. 1f exhibited almost similar microstructure as compared to the related papers.

  1. Comments on XPS data (lines 150-160) are unacceptably poor and completely lacking from a detailed comparison with pertaining literature works. The fitting in Fig. 2b is highly questionable and the top spectrum in Fig. 2d is almost at the noise level.

>> Thanks to the reviewer's suggestions, we have revised the description about XPS (lines 149-158). For the fitting of Fig. 2b, the peak fit was consistent with our previous literature (J. Mater. Res. 23 (2008) 2091). The Ref. (J. Mater. Res. 23 (2008) 2091) was cited in the revised manuscript. Moreover, as the reviewer comments "the top spectrum in Fig. 2d is almost at the noise level". It means that a very small amount of Ni is detected on the as-spun surface. This result is reasonable, because a large amounts of preferentially oxidized products (TiO2 and ZrO2) covered on Ni surface. Anyway, the top spectrum in Fig. 2d was just shown for comparison, which did not influence the XPS results of dealloyed samples.

  1. The supplementary video is almost useless.

>>We believe that the video is necessary as a support material in order to demonstrate the extremely fast degradation performance of this electrode.

  1. Sections 3.2 and 3.3 are very long-winded and not properly targeted to the presented experimental data.

>> Thanks to the reviewer's suggestions, the content has been modified in the revised manuscript.

  1. Fig. 3f: the equivalent circuit model used to fit the data is not reported.

>> Thanks to the reviewer's suggestions. The equivalent circuit model has been added.

  1. The role of chloride ions is not properly discussed and not adequately supported by chemical analyses on the working solutions to identify the real occurrence of the various species. In this regard, the mechanism section on page 11, is very speculative and not adequately supported, and Fig. 7 (which is also badly drawn) remains almost completely useless.

>> For chemical analysis of solutions, we regret that many tests could not be performed because of the COVID-19 epidemic. Among the active species, ·OH, ·O2- and 1O2 were detected by EPR tests. On the other hand, many reports demonstrated that active chlorine was produced by applying potential in the electrolyte containing NaCl (J. Alloys Compd. 882 (2021) 160700; Electrochim. Acta 263 (2018) 1–7; J. Mater. Sci. Technol. 92 (2021) 40–50). In this study, we used 0.1 M of Na2SO4 and NaCl as supporting electrolytes, respectively, to study the degradation performance of the electrodes, as shown in Figure 1. It can be clearly seen that the degradation performance of the electrode in the NaCl electrolyte is much greater than that in the Na2SO4 electrolyte. Therefore, we believe that NaCl plays an important role in the degradation process. In this work, under the applied 1.5 V potential condition, NaCl undergoes oxidation reaction and forms active chlorine species thus removing organic pollutants, which was proved in Fig. 6a-c.

  1. No chemico-physical analyses on the working materials after functional tests are presented and discussed.

>>We regret that many tests could not be performed because of the COVID-19 epidemic. The results of these related tests are difficult to be performed within 7 days revision.

  1. The Conclusions are very technical and not adequately supported by the data, and no future perspectives for the prosecution of the presented research activities can indeed be foreseen.

>> Thanks for your kind correction, we have modified the conclusions in the revised manuscript.

Reviewer 2 Report

This article (metals-2081243) the authors describes the “Performance and mechanism of nanoporous Ni@NiO composites for RhB ultrahigh electro-catalytic degradation”. Some quires need to be addressed, for that a major revision is required to enhance the quality of the manuscript.

1.      Explain the uniqueness of the prepared materials and synthesis methods, there are many reports already published on synthesis of nanoporous Ni@NiO composites with cost effective and simple methods with application of the prepared materials.

2.      Abstract needs to be enhanced by using the results values of the prepared materials.

3.      The Introduction part needs to be enhanced by adding a comparison table of previous reports and latest references.

10.1016/j.diamond.2022.109413; 10.1016/j.materresbull.2022.111911

4.      XRD studies should be indexed properly?

5.      Explain the XPS analysis properly and provide the fittings with proper resolutions as it is not visible the peak position/s.

6.      Explain the method used to calculate the pore size distribution.

7.      Provide the BET surface area studies for the synthesized materials.

8.      Explain the agglomeration of particles seems in SEM images.

9.      English of the manuscript needs to polished and rectified thoroughly.

10.  All the images/micrographs should have clear appearance with high resolution and uniform font size and scale markers.

Author Response

Reviewer #2: This article (metals-2081243) the authors describes the “Performance and mechanism of nanoporous Ni@NiO composites for RhB ultrahigh electro-catalytic degradation”. Some quires need to be addressed, for that a major revision is required to enhance the quality of the manuscript.

  1. Explain the uniqueness of the prepared materials and synthesis methods, there are many reports already published on synthesis of nanoporous Ni@NiO composites with cost effective and simple methods with application of the prepared materials.

>> A new concept, including the design of a Ni40Ti40Zr20 amorphous dealloying precursor and the construction of a unique self-standing and nanoporous structure in conjunction with facile one-step dealloying method, is established in this work. Although the nanoporous Ni@NiO composites have been studied in the fields of glucose sensors and supercapacitor. In this work, it is the first time that this material was applied to the field of electrochemical degradation of dyes and showed extremely high electrochemical performance, which broadens the scope of application of amorphous alloys and nanoporous metals.

  1. Abstract needs to be enhanced by using the results values of the prepared materials.

>> Thanks to the reviewer's suggestions. We have carefully modified the abstract in the revision.

  1. The Introduction part needs to be enhanced by adding a comparison table of previous reports and latest references.

10.1016/j.diamond.2022.109413; 10.1016/j.materresbull.2022.111911

>> We would like to thank the reviewer for providing the useful references which were cited in the revision.

  1. XRD studies should be indexed properly?

>> Thanks for your kind suggestion. We have carefully modified the XRD studies in the revision.

  1. Explain the XPS analysis properly and provide the fittings with proper resolutions as it is not visible the peak position/s.

>> Thanks to the reviewer's comments. We have optimized the XPS analysis.

  1. Explain the method used to calculate the pore size distribution.

>> Our previous studies (Applied Materials Today 19 (2020) 100539) reported that, the pore size distribution curve obtained from the N2 adsorption/desorption branches using the BJH method exhibited a narrower size distribution at 2.1 nm and a wider distribution at 7.6 nm, respectively, suggesting the electrode material contains a hierarchical pore structure. We cited the previous studies in the revised manuscript.

  1. Provide the BET surface area studies for the synthesized materials.

>> The BET methods were performed in our previous study (Applied Materials Today 19 (2020) 100539), the nanoporous Ni possessed high surface area of 100.3 m2 g-1, which is superior than most of nanoporous metals. In this work, the electrochemically active surface area for the electrodes were measured, as shown in Figure 3c, which is more relevant with the active sites.

  1. Explain the agglomeration of particles seems in SEM images.

>> No aggregation of particles is observed in the SEM image. During dealloying, the nanopores and the ligaments are formed due to the selective corrosion of Ti and Zr. The “agglomeration of particles” might be the ligaments. Notably, the ligaments are not individual ones, they interconnect together to maintain the self-standing properties of the as-dealloyed strips and that is the reason that we define them as the ligaments.

  1. English of the manuscript needs to polished and rectified thoroughly.

>> Thanks to the reviewer's suggestions. We have carefully checked the manuscript and the changes have been marked in the revision.

  1. All the images/micrographs should have clear appearance with high resolution and uniform font size and scale markers.

>> Thanks for your kind correction. The Ni-based materials possess the magnetic properties, resulting in the fuzzy SEM images. We further enlarged the SEM images in the revised Fig. 1b-e. Moreover, the font size and scale markers were also modified in the revision.

Reviewer 3 Report

The article entitled “Performance and mechanism of nanoporous Ni@NiO composites for RhB ultrahigh electro-catalytic degradation” is an article reporting the fabrication and performance analysis of nano-porous electrodes produced by de-alloying. The article looks well written to me. I am not expert on dye pollutant decontamination so my review concerns the material. The functional properties must be reviewed by another expert. In overall material production looks clear and the paper is acceptable to me (for the material production part).

I suggest adding a discussion of the pros and cons of the production process using HF acids and the gain between the production process which need a dangerous acid to be used and the depollution effect.

Author Response

Reviewer #3: The article entitled “Performance and mechanism of nanoporous Ni@NiO composites for RhB ultrahigh electro-catalytic degradation” is an article reporting the fabrication and performance analysis of nano-porous electrodes produced by de-alloying. The article looks well written to me. I am not expert on dye pollutant decontamination so my review concerns the material. The functional properties must be reviewed by another expert. In overall material production looks clear and the paper is acceptable to me (for the material production part).

  1. I suggest adding a discussion of the pros and cons of the production process using HF acids and the gain between the production process which need a dangerous acid to be used and the depollution effect.

>> A good question! HF is often used to etch glass. In this study, np-Ni@NiO electrodes with very high degradation properties were prepared by the dealloying method using a 0.05 M HF solution, while the industrialization promotion of our products suffers from the environmentally harmful residue. We will debug the composition of the alloy and use a more environmentally friendly preparation process in the future. In this manuscript, the discussion is mainly focused on degradation performance and degradation mechanism discussion of the self-standing np-Ni@NiO electrode.

Round 2

Reviewer 1 Report

-

Reviewer 2 Report

The manuscript in the form corrected is acceptable for publication!